

# YouTube as a source of information in cardiopulmonary resuscitation for 2020 AHA Resuscitation Guidelines

Merve Yazla[1], Tuba Şafak[1], Şakir Hakan Aksu[2], Kadiriye Savran[1], Rafet Fatih Aydogan[1], Mustafa Arslan[3], Abdullah Osman Koçak[4] and Burak Katipoğlu[1]

[1] Emergency Department, Ankara Etlik City Hospital, Ankara, Turkey
[2] Emergency Department, Samsun Education and Research Hospital, Samsun, Turkey
[3] Emergency Department, Mamak State Hospital, Ankara, Turkey
[4] Emergency Department, Balıkesir Atatürk City Hospital, Balıkesir, Turkey

Corresponding author
Tuba Şafak, tubasafak3@gmail.com

## ABSTRACT

**Background**. The Internet has transformed global information access, particularly through platforms like YouTube, which launched in 1995 and has since become the second largest search engine worldwide with over two billion monthly users. While YouTube offers extensive educational content, including health topics like cardiopulmonary resuscitation (CPR) and basic life support (BLS), it also poses risks due to potential misinformation. Our study focuses on evaluating the accuracy of CPR and BLS videos on YouTube according to the latest 2020 American Heart Association (AHA) guidelines. This research aims to highlight inconsistencies and provide insights into improving YouTube as a reliable educational resource for both lay rescuers and healthcare professionals.

**Methods**. In this cross-sectional observational study, English YouTube videos uploaded between October 21, 2020, and May 1, 2023, were searched using keywords related to CPR and basic life support. Videos were assessed for their source, duration, views, use of human or mannequin models, and mean assessment scores by two emergency medicine physicians. A third physician's opinion was sought in cases of disagreement. The first assessment evaluated video validity based on specified information criteria, while the second assessed their ability to convey advanced medical information aligned with the 2020 AHA guidelines.

**Results**. In this study, 201 English YouTube videos uploaded between October 21, 2020, and May 1, 2023, were evaluated based on search terms related to CPR and BLS, resulting in 95 videos meeting inclusion criteria after excluding 106 due to various reasons. Most included videos were from healthcare professionals (49.5%), followed by anonymous sources (29.5%) and official medical organizations (21.1%). Video durations ranged widely from 43 to 6,019 seconds, with an average of 692 seconds. Videos featuring mannequins predominated (91.6%), followed by those using human subjects (5.3%) or both (3.2%). Healthcare professional and official medical organization videos scoring significantly higher than those of unknown origin ($p = 0.001$). Video length did not correlate significantly with view counts, although shorter videos under 5 minutes tended to have higher average views.

**Discussion**. The results presented in this study demonstrated that English-language videos on YouTube related to BLS and CPR, throughout the study period, did not

conform to the 2020 AHA guidelines in terms of providing basic information for lay rescuers. Furthermore, healthcare professionals cannot obtain advanced medical knowledge through these videos. We recommend a professional oversight mechanism in health-related videos that does not tolerate such misinformation.

## INTRODUCTION

The Internet is a communication network that has revolutionized the access to and sharing of information (*Webster, 2001*). Particularly through devices such as smartphones and tablets, accessing information has become easier than ever. Based solely on search queries, YouTube is the second largest search engine in the world, following its parent company, Google. Approximately 700,000 h of video are watched every minute, and it has over two billion active users each month. Considering that one-third of these users watch 'how-to, educational, or instructional videos', YouTube also serves as a valuable source of educational material, capable of profoundly influencing people's behaviors and decisions (*Ranktracker, 2023*).

In their efforts to achieve informed health literacy, person are increasingly turning to the internet to better understand their medical conditions and treatments. However, with this growing opportunity comes the unfortunate potential for the spread of inaccurate and even harmful information. Online videos aimed at lay rescuer education, along with concerns regarding their quality and accuracy, have recently garnered significant attention. In a systematic review analyzing YouTube videos, it was revealed that while YouTube can provide high-quality health information, it can also host conflicting and misleading health-related content (*Madathil et al., 2015*). When these videos are not produced by healthcare professionals or reputable health organizations, the likelihood of spreading misinformation increases. Although analyses of YouTube videos containing health-related information are being published more frequently today, a standardized method or evaluation guideline has yet to be established (*Hancıet al., 2023*; *Kara et al., 2024*; *Erkin, Hancı& Özduran, 2023*; *Drozd, Couvillon & Suarez, 2018*).

It is well-known that the recognition of sudden cardiac arrest and the early initiation of cardiopulmonary resuscitation (CPR) significantly improve patient survival (*Mani, Annadurai & Danasekaran, 2015*). Therefore, it is anticipated that the more quality and accessible educational opportunities provided to the target audience, the higher the potential for improved survival rates (*Bhanji et al., 2010*). Since online health education and content are particularly important when it comes to basic life support (BLS) and CPR, YouTube can serve as a valuable source of information for lay rescuers and as an online educational tool for healthcare professionals.

It is essential that health-related content on YouTube undergoes a review process based on reliable, valid, and most importantly, up-to-date references to ensure its accuracy. While there have been studies analyzing BLS and CPR videos previously uploaded to

YouTube (*Yaylaci et al., 2014*; *Gezer et al., 2023*) our research did not identify any studies examining the compliance of English-language videos with the 2020 updated American Heart Association (AHA) guidelines. This study aims to evaluate the extent to which YouTube videos providing BLS and CPR training for lay rescuers align with the 2020 AHA guidelines. Additionally, it seeks to assess the usability of these videos as up-to-date and accurate educational material for healthcare professionals.

## MATERIALS & METHODS

In this cross-sectional and observational study, the YouTube (YouTube©, https://www.youtube.com; YouTube, LLC, San Bruno, CA, USA) website was searched for English-language videos uploaded between October 21, 2020, and May 1, 2023, using the following MeSH-compliant keywords: ''CPR'', ''cardiopulmonary resuscitation'', and ''basic life support''. In a previous study, YouTube videos had already been evaluated for BLS and CPR based on the 2015 AHA guidelines (*Katipoğlu et al., 2019*). This study, however, includes the evaluation of more recent videos that reflect updates in the guidelines. Therefore, the most recent AHA guidelines were used as the reference for this study. The date of October 21, 2020, was chosen as the starting point, as it marks the publication of the 2020 AHA guidelines.

Videos that met at least one of the exclusion criteria were not included in the study. These criteria were: 'non-medical content (advertisements, news, interviews), videos in languages other than English, pediatric CPR footage, live-action content lacking educational material, non-educational content such as comedy or entertainment, duplicate footage, content promoting CPR devices for commercial purposes, animal CPR footage, and duplicate videos'. Videos that were not educational in nature regarding BLS and CPR were excluded from the study.

For the videos included in the study, the following data were recorded: sources (official medical organizations: AHA, International Liaison Committee on Resuscitation (ILCOR), European Resuscitation Council (ERC); healthcare professionals and organizations: doctors, nurses, paramedical personnel, medical schools, hospitals; or unidentified sources), duration, view counts during the study period, model use (human, manikin, or both), and average scores (initial and subsequent evaluations). The sources of the videos were determined by examining the channel information where the video was published and verifying this by reviewing the official websites of the affiliated organizations.

The videos that met the inclusion criteria were reviewed by two emergency medicine specialists with similar professional experience (T.Ş. and Ş.H.A.). In cases of disagreement between the two physicians, the opinion of a third emergency medicine specialist (M.Y.) was sought.

The initial evaluation was conducted to assess whether YouTube could be used for BLS training for lay rescuers. The validity of the videos was assessed based on selected information considered important for the BLS algorithm. The scoring was summarized in Table 1 of the ''Part 3: Adult Basic and Advanced Life Support: 2020 American Heart Association Guidelines for Cardiopulmonary Resuscitation and Emergency Cardiovascular Care'' (Table 1) (*Panchal et al., 2020*).

**Table 1  Selected criteria from the basic life support algorithm used for video assessment.**

| Selected information from the basic life support algorithm | |
| --- | --- |
| (1) | Providing environmental safety |
| (2) | Control of patient unresponsiveness |
| (3) | Ensuring airway patency and assessing breathing |
| (4) | Activation of emergency medical system using mobile devices |
| (5) | C-A-B sequence |
| (6) | 30:2 chest compression |
| (7) | Correct localization of chest compression |
| (8) | Appropriate chest compression depth (5–6 cm) |
| (9) | Use of defibrillator |
| (10) | The rate of chest compressions should be 100–120 per minute. |

**Table 2  Selected innovations mentioned in the 2020 American Heart Association (AHA) guidelines used for video evaluation.**

| Selected innovations from the 2020 AHA guidelines | |
| --- | --- |
| (1) | Rapid initiation of CPR for individuals suspected of cardiac arrest (low risk of harm due to chest compressions) |
| (2) | Early administration of epinephrine in non-shockable rhythms |
| (3) | Administration of epinephrine if the first defibrillation attempt is unsuccessful in shockable rhythms |
| (4) | Use of auditory-visual devices is beneficial |
| (5) | ETCO2 > 10 mmHg |
| (6) | Dual sequential defibrillation is not beneficial |
| (7) | IV route is the primary route for drug administration; if IV access fails or is inappropriate, IO should be attempted |
| (8) | Administration of IM or IN naloxone |

Notes.

CPR, Cardiopulmonary Resuscitation; ETCO2, End-tidal carbon dioxide; IV, intravenous; IO, intraosseous; IM, intramuscular; IN, intranasal.

The second evaluation was conducted to assess the ability of YouTube videos to convey up-to-date medical information in BLS for healthcare professionals. For this reason, information related to significant updates in the 2020 AHA guidelines, which were not present in the previous guidelines (AHA 2015, Part 1: Executive Summary: 2015 American Heart Association Guidelines Update for Cardiopulmonary Resuscitation and Emergency Cardiovascular Care), was selected. The scoring criteria were summarized in Table 2, based on ''Part 1: Executive Summary: 2020 American Heart Association Guidelines for Cardiopulmonary Resuscitation and Emergency Cardiovascular Care, Adult Basic and Advanced Life Support: Important New, Updated, and Reaffirmed Recommendations'' section (Table 2) (*Neumar et al., 2015*; *Merchant et al., 2020*).

*Statistical analysis.* The statistical analysis of the research data was conducted using the SPSS 25.0 statistical software package (IBM, Armonk, NY, USA). The data were presented as means, standard deviations, medians, minimum and maximum values, percentages, and frequencies. The normal distribution of continuous variables was verified using the Shapiro–Wilk test. Analysis of variance (ANOVA) was employed to compare normally distributed continuous data across more than two groups. For non-normally distributed data, the Kruskal-Wallis test was used, followed by *post hoc* tests as necessary. Inter-rater

**Table 3  Number of excluded videos.**

| Exclusion reason | Number |
|---|---|
| Presence of non-medical content (advertisements, news, and interviews) | 20 |
| Non-English language | 33 |
| Pediatric cardiopulmonary resuscitation footage | 23 |
| Lack of educational content, presence of live-action footage (real-life videos) | 6 |
| Comedy and entertainment content | 4 |
| CPR device demonstrations | 17 |
| Duplicate footage | 19 |
| Footage of cardiopulmonary resuscitation on animals | 1 |
| Total | 106 |

reliability was assessed using Cohen's kappa. In all analyses, a $p$-value of less than 0.05 was considered statistically significant.

## RESULTS

A total of 201 English-language videos uploaded to YouTube between October 21, 2020, and May 1, 2023, were identified and evaluated using the search terms "CPR", "cardiopulmonary resuscitation", "basic life support", "chest compressions", and "advanced cardiac life support". Of these videos, 106 were excluded based on the exclusion criteria. The number of excluded videos and the reasons for their exclusion are detailed in Table 3. The inter-rater reliability, assessed using Cohen's kappa analysis, yielded a kappa value of 0.85, indicating strong agreement between the evaluators (Table 3).

The analysis included 95 videos that met the inclusion criteria. The majority of the videos were uploaded by healthcare professionals ($n = 47$, 49.5%), followed by videos uploaded by unidentified sources ($n = 28$, 29.5%), and videos uploaded by official medical organizations ($n = 20$, 21.1%).

The duration of the videos ranged from 43 to 6,019 s (mean, $692 \pm 858$ s; median, 401 s). Videos shorter than 5 min had the highest average number of views (mean, $317,986 \pm 1,563,382$; median, 2,376; minimum, 6; maximum, 9,666,179), followed by videos longer than 10 min (mean, 159,215; median, 8,800; minimum, 81; maximum, 1,980,996), while videos between 5 to 10 min had the lowest average number of views (mean, 70,801; median, 3,342; minimum, 86; maximum, 452,586). Although videos shorter than 5 min had the highest average view count, the number of views was not statistically correlated with video duration (Fig. 1).

Among the videos included in this study, 5.3% ($n = 5$) used human subjects, 91.6% ($n = 87$) utilized manikins, and 3.2% ($n = 3$) featured both human subjects and manikins together to demonstrate medical procedures.

Table 4 shows the number of videos containing each type of information. The average total scores for the videos, calculated based on the average scores, are presented in Table 5 (according to Table 1 and Table 2 criteria). Videos uploaded by healthcare professionals received the highest average total score ($9.49 \pm 2.47$), followed by those uploaded by official
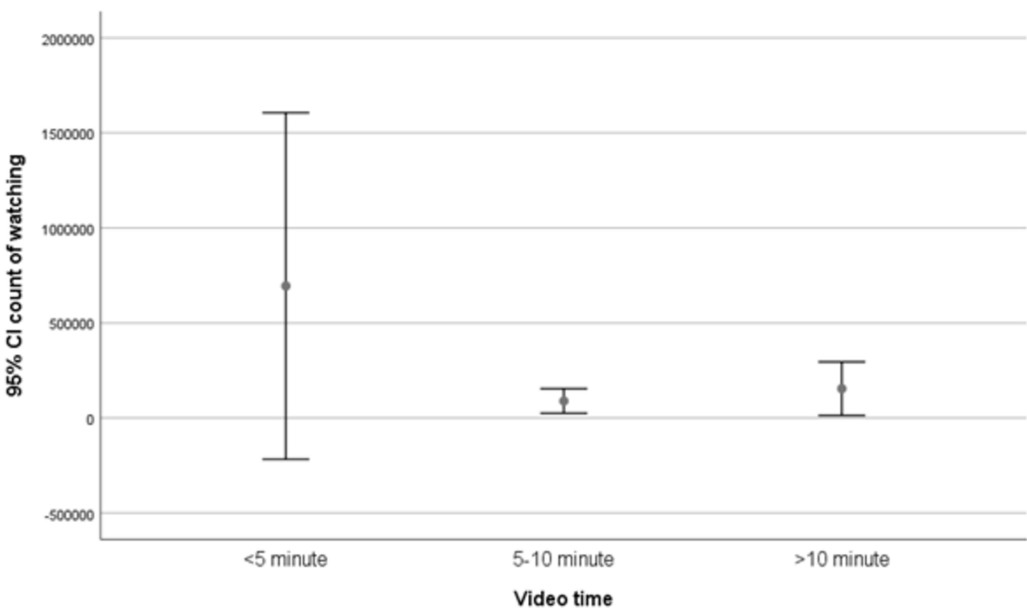

**Figure 1   Video durations and number of views.**

medical organizations (9.40 ± 2.50) and videos from unidentified sources (6.61 ± 2.94). The scores of videos uploaded by healthcare professionals and medical organizations were significantly higher than those of videos from unidentified sources, and this difference was found to be statistically significant ($p = 0.001$) (Table 4).

The average number of views, average video duration, and scores were presented according to the video source (Table 5). Based on the first evaluation and total score, the videos uploaded by healthcare professionals and official medical organizations had significantly higher scores compared to those from unidentified sources ($p = 0.001$). However, in the second evaluation, no statistically significant difference was found between the video sources (Table 5).

The relationship between the average number of views and scores was analyzed. According to the specified criteria:

- Videos that scored 1–2 points were viewed 2,442,912 ± 4,815,597 times (median, 52,687; minimum, 97; maximum, 9,666,179).
- Videos that scored 3–4 points were viewed 111,587 ± 262,925 times (median, 657; minimum, 7; maximum, 1,074,913).
- Videos that scored 5–6 points were viewed 60,432 ± 103,239 times (median, 21,281; minimum, 6; maximum, 369,761).
- Videos that scored 7–8 points were viewed 61,531 ± 115,419 times (median, 1,478; minimum, 74; maximum, 435,389).
- Videos that scored 9–10 points were viewed 132,609 ± 345,812 times (median, 11,155; minimum, 15; maximum, 1,980,996).

**Table 4** Number of videos containing information for each criterion.

| Information required in the study | Number of videos containing the required information (%) |
|---|---|
| Providing environmental safety | 51 (53,7%) |
| Control of patient unresponsiveness | 82 (86,3%) |
| Ensuring airway patency and assessing breathing | 48 (50,5%) |
| Activation of emergency medical system using mobile devices | 77 (81,1%) |
| C-A-B sequence | 58 (61,1%) |
| 30:2 chest compression | 76 (80%) |
| Correct localization of chest compression | 63 (66,3%) |
| Appropriate chest compression depth (5–6 cm) | 65 (68,4%) |
| Use of defibrillator | 64 (67,4%) |
| The rate of chest compressions should be 100-120 per minute. | 68 (71,6%) |
| Rapid initiation of CPR for individuals suspected of cardiac arrest (low risk of harm due to chest compressions) | 91 (95,8%) |
| Early administration of epinephrine in non-shockable rhythms | 16 (16,8%) |
| Administration of epinephrine if the first defibrillation attempt is unsuccessful in shockable rhythms | 29 (30,5%) |
| Use of auditory-visual devices is beneficial | 9 (9,5%) |
| ETCO2 > 10 mmHg | 8 (8,4%) |
| Dual sequential defibrillation is not beneficial | 0 |
| IV route is the primary route for drug administration; if IV access fails or is inappropriate, IO should be attempted | 11 (11,6%) |
| IM, IN naloxone | 2 (2,1%) |

**Notes.**
CPR, Cardiopulmonary Resuscitation; ETCO2, End-tidal carbon dioxide; IV, intravenous; IO, intraosseous; IM, intramuscular; IN, intranasal.

**Table 5** Average views, average video duration, and ratings by video source.

| | Uploaded by official medical organizations | Uploaded by healthcare professionals | Uploaded by unidentified sources | All videos |
|---|---|---|---|---|
| Average views | 118909 ± 185616 median: 7104 (15–633287) | 68769 ± 126253 median: 3712 (74–452586) | 704141 ± 2719303 median: 5257 (6–15000000) | 198526 ± 1013747 median: 4011 (6–9666179) |
| Scores based on first review[a] | 7,65 ± 2,62 median: 8,5 (1–10) | 7,60 ± 2,06 median: 8 (3–10) | 5,07 ± 2,61 median: 4 (0–10) | 6,86 ± 2,60 median: 8 (0–10) |
| Scores based on second evaluation[b] | 1,70 ± 1,08 median: 1,5 (0–4) | 1,87 ± 1,54 median: 1 (0–6) | 1,54 ± 1,50 median: 1 (0–6) | 1,74 ± 1,43 median: 1 (0–6) |
| Average total score | 9,40 ± 2,50 median: 10 (4–13) | 9,49 ± 2,47 median: 10 (4–15) | 6,61 ± 2,94 median: 5,5 (1–12) | 8,62 ± 2,91 median: 9 (1–15) |

**Notes.**
[a]Video scores according to criteria selected from basic cardiac life support algorithms*.
[b]Video scores based on what's new in 2020 American Heart Association guidelines.

There was no statistically significant relationship between the number of views and the score.

## DISCUSSION

We rely on information to prevent or mitigate health risks, and the quality of this information must be beyond dispute, especially when it concerns critical topics like BLS and CPR, where there is no room for error. It is unacceptable for outdated or inaccurate information to be readily available on the internet, where it is easily accessible to everyone. The unregulated and easily accessible nature of digital media increasingly challenges this principle. The results of this study indicate that the videos on YouTube providing BLS and CPR training for lay rescuers do not conform to the 2020 AHA guidelines. Furthermore, the findings suggest that these videos are inadequate as up-to-date educational material for healthcare professionals.

YouTube is not just a video-sharing platform; it is also a search engine. How search results are displayed depends on the algorithm the platform uses to rank a video's relevance to the search criteria. Patients use the internet to gather information on a wide range of medical topics, with chronic cardiovascular diseases being among the most frequently searched conditions. This emphasizes the need for patients and their families to be prepared by learning intervention techniques that could help save lives in the event of sudden cardiac arrest (*Diaz et al., 2002*). In this study, approximately 52% of the evaluated videos were excluded because they did not meet the inclusion criteria. In a similar study conducted in 2019, this exclusion rate was 88% (*Katipoğlu et al., 2019*). Despite the large volume of video content available, our research shows that even though the exclusion rate has decreased, one out of every two videos still fails to provide the desired information. This underscores the increasing use of YouTube as a tool for disseminating information about BLS and CPR, yet it also highlights the ongoing difficulty in accessing accurate and relevant content. Additionally, since these videos were not assessed for misinformation, it remains unclear to what extent users are accessing the excluded videos related to BLS and CPR and how much of the inaccurate information they may be applying in practice.

Popularity-based metrics on YouTube, such as view counts and likes, should not be considered indicators of the accuracy or timeliness of the content. Regardless of their specialty, every physician must be knowledgeable in basic life support and the approach to a patient in cardiac arrest. To protect the fundamental right to life, healthcare professionals should prioritize staying informed through current research rather than relying on popular content. Our study found that resuscitation videos designed by national and international associations with contributions from numerous academics are viewed less frequently than those uploaded by unidentified sources. However, according to both the first and second evaluations, videos uploaded by unidentified sources received the lowest scores. Therefore, associations should professionally manage their social media accounts, creating video content designed to become 'trending topics', thereby helping to disseminate accurate information and ensuring that these videos gain more visibility.

To evaluate the informational content of the videos, we selected the BLS algorithm and updates from the 2020 AHA guidelines. The median score for the first evaluation, which assessed videos that could be beneficial to lay rescuers, was 8/10. In a study by *Katipoğlu et al. (2019)* using the same evaluation criteria based on the 2015 AHA BLS guidelines,

this score was 4/10. The increase in scores in our study when evaluating according to the AHA BLS guidelines may be attributed to the higher proportion of videos from healthcare professionals and medical organizations (*Katipoğlu et al., 2019*). Although there has been an improvement, we still believe that the overall informational content remains low. This further supports the notion that YouTube is not a reliable source for medical and health-related information (*Osman et al., 2022*).

In the secondary evaluation, which focused on the content directed at healthcare professionals, the median score was found to be 1/8 (minimum: 0, maximum: 6). Similarly, in the study by *Katipoğlu et al. (2019)*, which evaluated content based on the 2015 AHA guidelines, the score was 0/10 (minimum: 0, maximum: 8). Although there appears to be a slight increase, these still notably low scores indicate that the videos contain incomplete and outdated advanced medical information for healthcare professionals. This raises the question: "Is the health information on the internet a goldmine or a minefield?" (*Tonsaker, Bartlett & Trpkov, 2014*). In a field like CPR, where there is no room for error, it is crucial to prevent both lay rescuers and healthcare professionals from stepping into a minefield of misinformation. To achieve this, expert reviews of medical videos should be taken into consideration, and their evaluation data should be integrated into the ranking algorithms of platforms like YouTube (*Osman et al., 2022*). Briefly label the good and the bad (*Tonsaker, Bartlett & Trpkov, 2014*).

When examining the first evaluation scores, the videos uploaded by official medical organizations and healthcare professionals (8.5; 8) had significantly higher scores than those uploaded by unidentified sources (4), consistent with previous reports ($p = 0.001$) (*Hancı et al., 2023*; *Elicabuk et al., 2016*; *Yaylaci et al., 2014*). However, in the second evaluation, a statistically significant difference was found among the video sources. While this indicates that useful videos may be available for lay rescuers, it also suggests that these videos should not be used as educational material for healthcare professionals. Despite 70% of the videos in the study being uploaded by healthcare professionals and official medical organizations, none of the videos included the critical update from the AHA 2020 guidelines regarding the lack of benefit from dual sequential defibrillation, one of the most important interventions for managing an arrest patient. Although the proportion of videos uploaded by healthcare professionals and official medical organizations has increased compared to the study by Katipoğlu et al., it is still evident that they fall short in keeping up with current information (*Katipoğlu et al., 2019*).

The scores obtained in the first evaluation did not show a significant relationship with the number of views for the videos included in this study. Although we might assume that higher-scoring videos would be viewed more frequently, previous studies have reported similar findings to ours (*Elicabuk et al., 2016*; *Yaylaci et al., 2014*; *Şaşmaz & Akça, 2018*). Ultimately, people cannot access the content without watching the video. If videos were rated by healthcare professionals before being published on the site and higher-rated videos were prioritized in the algorithm, it would allow viewers to decide whether to watch based on these ratings. This could, in turn, alter the relationship between the score and the number of viewers.

The main limitation of this study was that only English-language videos were analyzed cross-sectionally due to the dynamic nature of the online platform. Additionally, the inclusion and scoring criteria used in our study were subjective, and only the 2020 AHA Resuscitation Guidelines were considered for consistency. Other guidelines, such as those from ERC, Australian and New Zealand Committee on Resuscitation (ANZCOR), and ILCOR, were not evaluated in this study. This may have led to videos that are consistent with these other guidelines being considered inconsistent in our study.

## CONCLUSIONS

The findings presented in this study indicate that the English-language videos on BLS and CPR published on YouTube during the study period do not align with the 2020 AHA guidelines in providing basic information for lay rescuers. Furthermore, healthcare professionals cannot obtain advanced, up-to-date medical knowledge through these videos. In health-related videos, where there is no tolerance for misinformation, we recommend the creation of an algorithm, in collaboration between official medical organizations and YouTube, to ensure that videos appearing at the top of search results are those that provide accurate and reliable information.

### Funding
The authors received no funding for this work.

### Competing Interests
The authors declare there are no competing interests.

### Author Contributions
- Merve Yazla conceived and designed the experiments, prepared figures and/or tables, and approved the final draft.
- Tuba Şafak performed the experiments, prepared figures and/or tables, and approved the final draft.
- Şakir Hakan Aksu performed the experiments, authored or reviewed drafts of the article, and approved the final draft.
- Kadiriye Savran analyzed the data, prepared figures and/or tables, and approved the final draft.
- Rafet Fatih Aydogan analyzed the data, prepared figures and/or tables, and approved the final draft.
- Mustafa Arslan analyzed the data, authored or reviewed drafts of the article, and approved the final draft.
- Abdullah Osman Koçak conceived and designed the experiments, authored or reviewed drafts of the article, and approved the final draft.
- Burak Katipoğlu conceived and designed the experiments, authored or reviewed drafts of the article, and approved the final draft.

## Data Availability

The raw measurements are available in the Supplementary Files.

## Supplemental Information

Supplemental information for this article can be found online at http://dx.doi.org/10.7717/peerj.18344#supplemental-information.

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
