# Peer review of "YouTube as a source of information in cardiopulmonary resuscitation for 2020 AHA Resuscitation Guidelines"

_PeerJ, doi:10.7717/peerj.18344_

## Round 0.1 · original submission · Major Revisions

Thank you for submitting your manuscript titled "Assessment of CPR Educational Content on YouTube: A Comparison with 2020 AHA Guidelines." The reviewers have provided detailed and critical feedback, which highlights several key areas that need significant revision. The reviewers have identified a number of critical issues that must be addressed to improve the scientific validity and overall clarity of your study. Below, I have summarized the most critical points raised:

One of the primary concerns is the selection of the 2020 AHA guidelines as the sole standard for assessing the accuracy of the videos. Reviewer #1 pointed out that this approach does not consider the international context of the videos, which may be based on guidelines from other respected bodies like the European Resuscitation Council (ERC) or the Australian and New Zealand Committee on Resuscitation (ANZCOR). This limitation could lead to biased assessments of otherwise accurate educational content. I recommend reviewing and possibly revising your scoring system and methodology to account for these international variations, or alternatively, providing a strong justification for the exclusive use of AHA guidelines.

The methodology for scoring videos also raised significant concerns. Specifically, the differentiation between basic and advanced life support content was not sufficiently clear, which may have led to inappropriate evaluations, particularly in the scoring of basic life support videos. Reviewer #1 suggested separating the assessment and reporting of basic and advanced life support videos to avoid conflation and potential under-evaluation. Additionally, the criteria for scoring appear to omit key recommendations while including others that may not be universally applicable, such as naloxone administration. A more transparent and methodologically sound process for selecting and applying scoring criteria is necessary.

Reviewer #2 noted that many statements in the manuscript, particularly in the introduction and discussion sections, lack appropriate citations. Strengthening these sections with relevant literature will improve the academic rigor of your manuscript. Additionally, all references to the 2020 AHA guidelines should be accurately cited in the reference list.

Both reviewers emphasized the need for clearer and more precise language throughout the manuscript. This includes avoiding informal phrasing and ensuring that all abbreviations are fully spelled out when first introduced. Reviewer #2 also suggested that the manuscript could benefit from a more formal tone to align with academic standards.

The discussion section needs to be expanded to more thoroughly explore the implications of your findings. Reviewer #2 suggested that you discuss potential public health risks associated with misinformation in CPR videos and offer actionable strategies for mitigating these risks, such as recommendations for YouTube’s algorithm to prioritize accurate medical content.

There were concerns about the validity of the findings, particularly due to the methodological issues mentioned above. Reviewer #2 also recommended that inter-rater reliability be assessed and reported using a measure like Cohen's kappa, which would enhance the transparency and rigor of your evaluation process.

Given these critical issues, the manuscript will require substantial revisions. Once these revisions have been made, we would be happy to reconsider your manuscript for publication.

Reviewer 1 ·

Basic reporting

Thank you for the opportunity to review the manuscript by Yazla et al. that investigated the quality of health information related to cardiopulmonary resuscitation (CPR) available on YouTube by comparing it to the latest AHA guidelines on CPR.

Overall, the article is well written. The authors use several abbreviations in the manuscript which should be explained when first used in the text. I would suggest avoiding all abbreviations in the title and abstract. If an abbreviation cannot be avoided it should be explained when used for the first time in the abstract.

The introduction helps the reader to understand the authors perspective and study aim but is in many parts lengthy describing common knowledge that does not strengthen the authors narrative. I would suggest that they shorten the first paragraph (lines 49-64) to allow for a more concise introduction to their study.

The authors use the terms ‘patients’ and ‘lay rescuers’ seemingly interchangeably in the manuscript when they want to refer to members of the public who are not healthcare professional. I suggest using uniform and inclusive terminology to describe this group.

The authors refer to the 2020 American Heart Association (AHA) guidelines throughout the manuscript and use them as the standard to assess all reviewed material. However, none of the 2020 AHA guidelines are cited – only the 2010 AHA guidelines on education, implementation and teams (Part 16) by Bhanji et al. can be found in the reference list. I suggest that the authors review the 2020 AHA guidelines carefully and reference those guideline parts relevant to this manuscript.

The authors supplied their raw data as an .sav file which is the file format used in their statistical program (SPSS). While the format can also be imported using R or other open access tools, the authors may want to consider a more accessible file format.

Experimental design

I believe the aim to assess the accuracy of publicly available educational material for CPR on YouTube to be generally worthwhile to investigate and that the study may be relevant to the journal. However, the authors do not define a specific knowledge gap and do not sufficiently explain how their investigation may add to the existing knowledge cited in their paper (lines 82-86).

The authors aimed to identify discrepancies between educational videos from any source publicly available on YouTube and the 2020 AHA Guidelines for CPR. However, there appear to be major issues with the underlying assumption of the study aim and with the methodology chosen to address this aim. Please find my concerns explained below:

1. The authors chose to define the 2020 AHA guidelines as the standard for accurate information. While this may seem appropriate at first, they then include any video on YouTube without considering in which international context the video was created and what the target audience was. This means that videos based on other established guideline bodies that publish guidelines in English (such as the CPR guidelines produced by the European Resuscitation Council (ERC) or Australian and New Zealand Committee on Resuscitation (ANZCOR)) were assessed based on AHA guidelines. This disregards the fact that clinical practice guidelines dealing with the same healthcare topic might deviate from each other due to contextualized recommendations. It is known that AHA and ERC guidelines differ on key recommendations as guideline authors consider medical practice, patient preferences as well as scientific evidence for the community they serve. If the goal was to assess the broadest international consensus on resuscitation science, the authors could have considered to use the annual Consensus on Science and Treatment Recommendations (CoSTRs) issued by the International Liaison Committee on Resuscitation (ILCOR) as a reference standard. However, given the international composition of creators and viewers active on a platform like YouTube, it seems inappropriate to single out one guideline body’s recommendation and deem all content created based on other guideline bodies as inaccurate in cases where such guideline bodies disagree. I suggest that the authors review and potentially revise their scoring system and methodological approach to ensure that such discrepancies between guidelines do not lead to false assessments of otherwise accurate educational material.

2. The authors chose to assess basic and advanced life support content in this study using two scoring systems allowing for 10 and 8 points per respective type of life support. The total score of 18 points could not apply to basic life support content as this would not be appropriate to teach to non-qualified personnel or lay persons. However, the authors do not specific if videos addressing basic life support were only assessed for the items that apply to basic life support. Except for Table 5 (primary score row), the authors appear to report scores in the context of an 18-point maximum (lines 36-29; line 134; Table 4). This apparent misconception may have affected the calculation of subsequent results and appears to introduce a serious systematic under-evaluation of basic life support videos. I suggest that the authors rate and report basic and advanced life support videos separately to avoid any inappropriate conflation.

3. The authors provide little explanation how they decided on the scoring criteria used to assess guideline-compliant videos. When reviewing the tables in which they are specified, it appears that these were not exhaustive and omit key recommendations such as limiting chest compression pause durations. The tables also include criteria which may not always be relevant to all cardiac arrest scenarios. For example, Table 1 appears to only refer to out-of-hospital cardiac arrest (OHCA) as it requires a call for emergency medical services using a mobile device – this recommendation would likely not apply in the in-hospital setting and would mean that the authors systematically downgraded educational videos outside of the bystander CPR OHCA setting. Another potentially problematic example is the choice of including the administration of naloxone in Table 2 as a required point. Naloxone administration is relevant to opioid-associated emergencies and is recommended in this context. This would mean that more general advanced life support educational material not highlighting the naloxone administration would have been inappropriately downgraded by the authors.
Unfortunately, the approach makes it unlikely that any of the results based on the authors’ scoring system can inform a reader as they introduce bias as some key recommendations are scored while others are left out and because context specific interventions are not considered adequality.
I suggest that the authors review and revise their methodological approach to assessing and scoring the videos. A transparent and more complex process of selecting scoring criteria is likely required. Such a scoring system should be based on an established method or tool. A potential starting point may be performance assessments used during simulation-based CPR training of healthcare providers.

4. The authors aimed to describe the source of uploaded videos. However, the do not specify how or if they verified that an account providing content was in fact a healthcare provider or official account of a medical association. Given that the authors were concerned about misinformation and interested in verification of content it would be valuable to assess the authenticity of the source to allow for a reliable interpretation of their results.

Validity of the findings

As my comments on methodology affect the validity of findings, I would suggest that the authors review the section ‘Experimental Design’.

The authors may consider reporting medians with interquartile ranges.

When formulating conclusions, I would argue for the formulation of a suggestion rather than a recommendation, which usually requires broader scientific consensus and more robust evidence.

Additional comments

The limitations section of the article is rather brief and would benefit from elaboration as there appear to be several potential limitations to the study design and subsequent interpretation.

Overall, it may be helpful to provide some of the results as graphs. For example, the number of views (lines 124-130) may be more intuitively understood when represented visually.

Reviewer 2 ·

Basic reporting

The study evaluates the accuracy and quality of YouTube videos related to CPR and Basic Life Support (BLS) in accordance with the 2020 American Heart Association (AHA) guidelines. Using a cross-sectional observational design, the authors analyze a wide range of English-language videos uploaded between 2020 and 2023, assessing their ability to provide correct medical information for both laypersons and healthcare professionals. The study aims to identify gaps in content and provide insights into improving YouTube as a reliable educational resource for critical life-saving techniques.

The manuscript addresses a highly relevant and timely issue. This focus is essential given the increasing reliance on digital platforms for health education. The cross-sectional study design is well-suited to the research question and allows for a broad analysis of video content across multiple sources. This approach provides valuable insights into the quality of publicly available medical information and highlights the need for greater accuracy and oversight in online health education resources.

Despite the clear relevance and importance of the topic, I have a few constructive comments and suggestions to help further strengthen the manuscript.

1. Lack of Citations
Many statements in the manuscript lack appropriate citations, especially where evidence or prior research is necessary to support the claims. For instance, the introduction contains general assertions, such as "YouTube also poses risks due to potential misinformation," without citing relevant research. Incorporating more references to the existing literature, particularly in the introduction and discussion sections, would provide greater academic rigor and strengthen the manuscript. This would also improve the clarity of arguments presented.

2. Language and Professional Tone
The manuscript’s language could be made more formal and precise to align with academic standards. Informal phrasing such as "We wanted to examine" detracts from the professional tone expected in a high-quality scientific manuscript. More appropriate phrasing would be, for example, "This study aims to assess." Additionally, the manuscript contains long and sometimes convoluted sentences that could be simplified for better clarity. Polishing the manuscript with a focus on concise, unambiguous language would greatly enhance both readability and professionalism.

3. Use of Abbreviations
To improve the manuscript’s clarity, I recommend ensuring that all abbreviations are fully spelled out the first time they are introduced. This will make the manuscript more accessible to readers who may not be familiar with certain terms or acronyms, improving its readability for a wider audience.

Experimental design

4. Search Strategy
The manuscript lacks clarity on whether the search terms were applied in a single comprehensive search or multiple individual searches. This distinction is important because a single search could limit the diversity of the sample. If only one search was performed, I suggest conducting multiple searches to capture a broader range of videos. Additionally, if multiple searches were conducted, it would be helpful to provide details on how many videos each search term contributed to the final dataset, enhancing the transparency of the methodology.

5. Evaluation of Inter-Rater Agreement
The manuscript does not report how much agreement existed between the reviewers. For greater transparency, I recommend calculating and including a Cohen's kappa coefficient. This statistical measure would quantify inter-rater reliability and help demonstrate the consistency of the evaluation process. Including this information would strengthen the manuscript's methodological rigor.

6. Inclusion and Exclusion Criteria
The inclusion and exclusion criteria for selecting videos are not explicitly stated in the methods section, making it difficult to assess the replicability of the study. For instance, it is unclear why certain types of videos, such as those demonstrating CPR devices, were excluded. Providing a detailed rationale for the inclusion and exclusion criteria would ensure transparency and enable readers to fully understand how the video selection process was carried out.

Validity of the findings

7. Justification of the Time Frame
The selected time frame of 2020–2023 requires further clarification. If the study’s goal is to assess whether YouTube videos adhere to the 2020 AHA guidelines, it would be useful to explain why older videos are not included, given that they might also be contributing to misinformation. Alternatively, if the focus is strictly on newer videos that reflect updates to the guidelines, this should be explicitly stated in the study’s aim. Providing a more detailed justification for this time frame would help solidify the study's purpose and ensure that the findings align with the research question.

8. Weak Discussion
The discussion section does not fully explore the implications of the study’s findings. There is a missed opportunity to provide a more comprehensive analysis of the potential public health risks posed by misinformation in CPR videos, particularly in terms of the harm this may cause to both laypersons and healthcare professionals. While the authors suggest professional oversight mechanisms, there is little detail on how such systems could be realistically implemented. I suggest broadening the discussion to include actionable strategies, such as how YouTube's algorithms might be adapted to prioritize accurate medical content. Additionally, the discussion could benefit from more concrete recommendations for future research, such as studying the impact of media literacy on the public’s ability to discern reliable sources of health information.

Additional comments

9. Definition of Anonymous Creators
The manuscript refers to videos uploaded by “anonymous” creators but does not provide a clear explanation of what this term entails. I recommend elaborating on this category, especially in terms of what distinguishes anonymous creators from healthcare professionals and official medical organizations. A more detailed explanation will help clarify the classification of the videos and provide a better understanding of the reliability of content based on the source.

---

## Round 0.2 · accepted · Accept

Dear Authors,

I would like to congratulate you on the new version of your manuscript. The revisions have significantly improved the clarity and overall quality of the paper. All concerns raised in the previous review have been addressed thoroughly and effectively.

In my view, the manuscript is now ready for publication. I appreciate your hard work and the attention to detail that has resulted in this high-quality submission.

Best regards,

Reviewer 2 ·

Basic reporting

No comments

Experimental design

No comments

Validity of the findings

No comments

Additional comments

The authors have adequately revised their paper in response to the reviewers' comments, and I recommend that the article be published.